# miR-126-3p and miR-21-5p as Hallmarks of Bio-Positive Ageing; Correlation Analysis and Machine Learning Prediction in Young to Ultra-Centenarian Sicilian Population

**DOI:** 10.3390/cells11091505

**Published:** 2022-04-30

**Authors:** Giulia Accardi, Filippa Bono, Giuseppe Cammarata, Anna Aiello, Maria Trinidad Herrero, Riccardo Alessandro, Giuseppa Augello, Ciriaco Carru, Paolo Colomba, Maria Assunta Costa, Immaculata De Vivo, Mattia Emanuela Ligotti, Alessia Lo Curto, Rosa Passantino, Simona Taverna, Carmela Zizzo, Giovanni Duro, Calogero Caruso, Giuseppina Candore

**Affiliations:** 1Laboratory of Immunopathology and Immunosenescence, Department of Biomedicine, Neurosciences and Advanced Technologies, University of Palermo, 90134 Palermo, Italy; giulia.accardi@unipa.it (G.A.); mattiaemanuela.ligotti@unipa.it (M.E.L.); calogero.caruso@unipa.it (C.C.); giuseppina.candore@unipa.it (G.C.); 2Department of Economics, Business and Statistics, University of Palermo, Viale delle Scienze, Building N. 13, 90128 Palermo, Italy; filippa.bono@unipa.it; 3Institute for Biomedical Research and Innovation, National Research Council, 90146 Palermo, Italy; giuseppe.cammarata@cnr.it (G.C.); riccardo.alessandro@unipa.it (R.A.); giuseppa.augello@irib.cnr.it (G.A.); paolo.colomba@irib.cnr.it (P.C.); alessia.l_86@hotmail.it (A.L.C.); simona.taverna@cnr.it (S.T.); carmela.zizzo@irib.cnr.it (C.Z.); giovanni.duro@irib.cnr.it (G.D.); 4Institute of Translational Pharmacology, National Research Council, 90146 Palermo, Italy; 5Clinical and Experimental Neuroscience, Institute for Aging Research, Biomedical Institute for Bio-Health Research of Murcia, School of Medicine, University of Murcia, Campus Mare Nostrum, 30100 Murcia, Spain; mtherrer@um.es; 6Section of Biology and Genetics, Department of Biomedicine, Neuroscience and Advanced Diagnostics, University of Palermo, 90127 Palermo, Italy; 7Department of Biomedical Sciences, University of Sassari, 07100 Sassari, Italy; carru@uniss.it; 8Institute of Byophysics, National Research Council, 90146 Palermo, Italy; mariaassunta.costa@cnr.it (M.A.C.); rosa.passantino@ibf.cnr.it (R.P.); 9Department of Epidemiology, Harvard T.H. Chan School of Public Health, Boston, MA 021382, USA; nhidv@channing.harvard.edu

**Keywords:** ageing, inflamm-ageing, endothelial senescence, longevity, miRNAs

## Abstract

Human ageing can be characterized by a profile of circulating microRNAs (miRNAs), which are potentially predictors of biological age. They can be used as a biomarker of risk for age-related inflammatory outcomes, and senescent endothelial cells (ECs) have emerged as a possible source of circulating miRNAs. In this paper, a panel of four circulating miRNAs including miR-146a-5p, miR-126-3p, miR-21-5p, and miR-181a-5p, involved in several pathways related to inflammation, and ECs senescence that seem to be characteristic of the healthy ageing phenotype. The circulating levels of these miRNAs were determined in 78 healthy subjects aged between 22 to 111 years. Contextually, extracellular miR-146a-5p, miR-126-3p, miR-21-5p, and miR-181a-5p levels were measured in human ECs in vitro model, undergoing senescence. We found that the levels of the four miRNAs, using ex vivo and in vitro models, progressively increase with age, apart from ultra-centenarians that showed levels comparable to those measured in young individuals. Our results contribute to the development of knowledge regarding the identification of miRNAs as biomarkers of successful and unsuccessful ageing. Indeed, they might have diagnostic/prognostic relevance for age-related diseases.

## 1. Introduction

microRNAs (miRNAs) are single-stranded and non-coding RNA molecules of 21–23 nucleotides that negatively regulate gene expression, which are involved in a wide range of physiological and pathological conditions. They have been observed to freely circulate in human body fluids such as blood, saliva, cerebrospinal and amniotic fluids, breast milk, and urine or in extracellular vesicles (EVs), and are selectively packaged and secreted by cells [1,2,3]. Recent findings demonstrate that senescent endothelial cells (ECs) are a source of circulating miRNAs, so they have been suggested as potential targets to counteract the ageing process [1,4]. The modulation of miRNAs transcription or their circulating levels have been also associated with age-related diseases [4,5].

The studies relating miRNA profiles to human longevity are relatively few. Since miRNAs pathways are evolutionary conserved, it is possible to formulate deep insights into the topic thanks to experimental models. Studies on *C. elegans* and mammals have demonstrated the role of miRNAs in modulating ageing, longevity, and cellular senescence. In *C. elegans* the modulation of miRNA maturation and functions can modify the lifespan and altered longevity phenotypes [6,7].

Changes in circulating miRNA levels could be considered a measure of biological ageing; accordingly, some miRNAs are proposed as new potential biomarkers of ageing and age-related diseases [4,8,9,10,11,12,13,14,15]. Recent findings indicate that several miRNAs are involved in inflamm-ageing, an ageing-related state characterized by systemic chronic low-grade inflammation that in turn supports a biological-background inducing predisposition to age-related diseases.

Specific miRNAs such as miR-126-3p, miR-21-5p, miR-146a-5p, and miR-181a-5p are widely recognised as senescence-associated miRNAs (SA-miRNAs) and regulate the expression of genes involved in ageing and longevity [16,17].

miR-126-3p is one of the most described SA-miRNAs and is considered as an endothelial cell-specific miRNA that regulates vascular integrity and angiogenesis [18]. Circulating miR-126-3p, released by ECs, exerts protective mechanisms against vascular endothelial dysfunctions involved in age-related vascular diseases [19,20].

miR-21 is an inflamma-miRNA that orchestrates the tuning of the inflammatory response. It can be considered as an miRNA that is able to modulate the “switch on/off” of inflammation at appropriate times [21]. Since ageing is associated with the alteration of redox signalling and increased circulating inflammatory cytokines, miR-21 circulating levels could be useful biomarkers of age-related diseases and their complications [22].

miR-146a is also considered an inflamm-miRNA and imparts several beneficial effects for healthy ageing. It was demonstrated that circulating miR-146a levels decline in healthy older people. Both miR-146a and miR-21 are able to track physiological and pathological ageing trajectories, so the monitoring of these miRNAs can contribute to the promotion of healthy ageing and prevent or postpone age-related diseases onset [23].

miR-181a-5p is known as ageing-related mito-miRNA since it can modulate mitochondrial activity. Its modulation could mediate the loss of mitochondrial integrity and function in ageing cells, contributing to the inflammatory response [24].

In the present study, the levels of miR-126-3p, miR-21-5p, miR-146a-5p, and miR-181a-5p were analyzed in the plasma samples of different age-cohorts, from young to ultra-centenarians, of a population from Western Sicily.

The aim of the paper was to identify specific miRNAs as potential biomarkers of healthy ageing and their correlation with age and sex as well as with smoking habits, and other biological parameters. miRNA levels have also been evaluated in the conditioned media of human umbilical vein endothelial cells (HUVECs) undergoing replicative senescence.

Furthermore, since some specific biological features seem to be age-associated, it was possible to find the relation between these features and people’s age using machine learning (ML) techniques. ML is a branch of Artificial Intelligence focused on imitating the way human beings are capable of learning by taking experimental data and examples as inputs to infer a specific result or output [25]. ML often builds mathematical models based on sample data to make predictions without being explicitly programmed for such purposes. In general, ML models are considered as black boxes where inputs are transformed into outputs by combining such inputs with a set of adjusted data obtained as a result of a training process [26]. Although it provides many benefits in different areas of application, the lack of interpretability could be a problem in some circumstances. In the context of this work, the age prediction problem is also addressed by considering several biological features, since it is interesting to study which factors affect the population from young to ultra-centenarians. Finding such relations in an interpretable manner is important; therefore, it is necessary to use ML techniques with a high degree of interpretability. Consequently, a decision-tree classifier has been selected for this purpose, since such an ML method presents a high rate of accuracy, great robustness and the results are very interpretable as long as they are short.

## 2. Materials and Methods

### 2.1. Study Design, Participants, and Anamnestic Data

The participants were recruited from June 2017 to March 2020 within the project “Discovery of molecular and genetic/epigenetic signatures underlying resistance to age-related diseases and comorbidities (DESIGN, 20157ATSLF)”, funded by the Italian Ministry of Education, University and Research. The Ethics Committee of Palermo University Hospital (Palermo, Sicily, Italy) approved the study protocol (Nutrition and Longevity, No. 032017). The study was conducted in accordance with the Declaration of Helsinki and its amendments. Study participants (or their caregivers) gave their written informed consent prior to enrolment.

All recruited subjects were Sicilians aged between 22 and 111 years. We excluded people with chronic invalidating diseases, such as neoplastic and autoimmune diseases, as well as with acute diseases, such as infectious diseases, and individuals with severe dementia. To respect privacy, all donors were identified with an alphanumeric code and the data were managed using a database accessible exclusively by researchers involved in the project. A team composed of demographers, statistics, biologists, and physicians, from University of Palermo, disseminated a detailed questionnaire to collect demographic, clinical, and anamnestic data of interest from participants as well as functional and cognitive information. The enrolment was conducted at University of Palermo for young adults, adults, and older adults, using social networks and word of mouth for recruitment, whereas it was conducted at home for the ultra-centenarians. For more information about the recruited population, please see [27].

A total of 78 healthy donors (females: 42; males: 36) were randomly selected from this large database. The population was divided in four age groups, i.e., young adults (22–50 years, *n* = 19), adults (51–70 years, *n* = 28), older adults (71–99 years, *n* = 20), and ultra-centenarians (100–111 years, n = 11).

The age, gender, smoking habits and some anthropometric, hematological, haematochemical, oxidative, and molecular parameters of the recruited subjects are reported in Appendix A.

### 2.2. Plasma Sample Acquisition and RNA Isolation

Overnight fasting blood samples were obtained in the morning and processed as previously reported [27]. Total RNA was extracted and purified from plasma, using an miRNeasy^®^Mini kit (Qiagen, CA, cat.No. 217004), according to standard protocol. The RNA concentration was assessed, using the RNA Nano 6000 Assay Kit of the Agilent Bio-analyzer 2100 System (Agilent Technologies, Palo Alto, CA, USA). RNA quality was assessed with the Eppendorf biophotometer D30 (Eppendorf, Hamburg, Germany). For this study, we used only RNA with a ratio of A260/280 from 1.9 to 2.

### 2.3. TaqMan RT-qPCR miRNA Assays

The isolated miRNAs were retro-transcripted using an miScript Single Cell qPCR kit (Qiagen, Hilden, Germany), according to the manufacturer’s protocol. The expression levels of miRNAs were evaluated with a SYBR green-based Real-Time quantitative PCR (RT-qPCR), using Step one plus (Applied Biosystem, Waltham, MA, USA). For the amplification, we used an miScript SYBR green PCR kit (Qiagen, Hilden, Germany) according to the manufacturer’s protocol. The 20 μL PCR mixture included 2 μL of reverse transcription product, 10 μL of QuantiTect SYBR Green PCR Master Mix, 2 μL of miScript Universal Primer, 4 μL of RNase-free water and 2 μL of Primer Assay specific for each microRNAs (miScript Primer Assays miR-21-5p Lot. N°201803230019, miR-126-3p Lot. N°20150817013s, miR-146a-5p Lot. N°201709120071, miR-181a-5p Lot. N°20161222018 Qiagen, Hilden, Germany). The reaction mixtures were incubated at 95 °C for 15 min, followed by 40 amplification cycles of 94 °C for 15 s, 55 °C for 30 s, and 70 °C for 30 s. Triplicate samples and inter-assay controls were used. Therefore, for the normalization of RT-qPCR data, using the 2-DCT method, we used miR-30a (miScript Primer Assay Lot. N°201709270012 Qiagen, Hilden, Germany). Linear fold changes were calculated and plotted on scatter plots using Prism (GraphPad Prism Software, San Diego, CA, USA).

### 2.4. Cell Cultures

HUVECs (ATCC^®^CRL-1730™, Manassas, VA, USA) were grown in Medium 200 (GIBCO) supplemented with hydrocortisone (1 µg/mL), human epidermal growth factor (10 ng/mL), basic fibroblast growth factor (3 ng/mL), heparin (10 µg/mL), gentamicin (10 µg/mL), amphotericin (0.25 µg/mL), and fetal bovine serum (2% *v*/*v*), at 37 °C, in a 5% CO_2_ atmosphere, at 95% humidity. For cell-replicative senescence, first passage cryopreserved HUVECs were grown and serially passed until they reached senescence [28]. The number of population doublings (PDs) was calculated by using the following formula: PD = [ln (number of cells harvested)–ln (number of cells seeded)]/ln2. For the experiments, cells were used at different cumulative population doublings (CPDs) which is the sum of all PD. Cells studied in early passage (CPD < 20) were regarded as young cells, whereas those that have been passed more times were regarded as intermediate-age (CPD < 32) or old, i.e., senescent (CPD > 56) endothelial cells (ECs).

### 2.5. Collection of Cell-Conditioned Media and miRNA Isolation

The conditioned media, containing all factors secreted by the cell line, were harvested from the cultures, centrifuged for 4 min at 300× *g* at 4 °C × 10 min, and stored at −20 °C until required for extraction. Total RNA was extracted by phenol/guanidine-based lysis of samples and a silica-membrane-column-based purification using an miRNeasy^®^ Mini kit (Qiagen, CA, cat.No. 217004), according to standard protocol. The RNA concentration was assessed using RNANano 6000 Assay Kit of the Agilent Bio-analyzer 2100 System (Agilent Technologies, Palo Alto, CA, USA).

### 2.6. Senescence-Associated β-Galactosidase Staining

HUVECs were grown in a 4-well chamber slide (Nunc™ Lab-Tek™ II, Thermo Fisher Scientific, Fremont, CA, USA), at a density of 6 × 10^4^ cells/well, in 500 µL of culture medium, for 24 h. At the end of the different times of doubling, HUVECs were fixed and stained for galactosidase activity using a Senescence Cell Staining kit according to the manufacturer instructions (Sigma-Aldrich, St. Louis, MO, USA). The percentage of senescence-associate gal positive cells was determined by counting the number of blue cells within a sample, using a Zeiss Axioskop microscope (Carl Zeiss, Göttingen, Germany) with an X20 lens. Ten random fields were photographed for each passage, and the percentage of SA-β-gal-positive cells were calculated.

### 2.7. MTS Assay

Cell viability was evaluated by using CellTiter 96 Aqueous One Solution Cell Proliferation Assay (Promega, Madison, WI, USA), a colorimetric method for determining the number of viable cells in proliferation, according to manufacturer instructions.

### 2.8. miRNA Targeted Gene Prediction and KEGG Pathway Analyses by miRWalk

We utilized miRWalk, an miRNA target gene prediction database, to select predicted and validated targets, and to analyze enriched KEGG pathways [29]. The miRWalk prediction database integrated other programs including miRanda, miRDB, RNA22, and Targetscan. Moreover, KEGG is an exhaustive database for the functional interpretation and practical application of genomic information and integrates macromolecular datasets from genome sequencing and other high-throughput experimental techniques [30,31].

### 2.9. Statistical Analyses of miRNA Levels

All calculations were made by using Stata 16.1 Software (StataCorp, College Station, TX, USA). Descriptive statistics were calculated for all the data considered in the study. Data from in vitro experiments were expressed as means and standard error of the means from at least three independent experiments. Comparison among multiple groups, in data with normal distribution and with a homogeneity of variance, was analysed by one-way analysis of variance (ANOVA). Kruskal–Wallis and Dunn tests with Bonferroni correction were considered to compare groups when data were not normally distributed and in the presence of heterogeneity. A robust multiple regression analysis was considered to look at the relationship between miRNAs and some individual and lifestyle characteristics of sample units. The robust estimation method corrects the standard errors of estimated coefficients in the presence of heteroskedasticity. The Wald test based on the robustly estimated variance matrix was considered to evaluate the significance of coefficients and R^2^ as a goodness-of-fit statistic. In this study, statistical significance was assumed when *p* < 0.05.

### 2.10. ML Techniques

The dataset was randomly divided into training and test sets. Age prediction was carried out to see which factors affect the population from young to ultra-centenarians [32]. A training and evaluation ratio of 7:3 was used. The ML algorithm used was the decision tree classifier. The decision tree algorithm belongs to the family of supervised ML algorithms [26]. The decision tree construction process consists of the selection of characteristics. Each attribute or characteristic is calculated following certain standards that allow for the selection of the most important characteristic that are characteristic of the partition samples each time. One of the main advantages of this algorithm is its high classification precision and great robustness [33,34]. To obtain reliable data, all data provided in the database were revised. Once each optimal model was found, they were fitted to the entire training set and tested on the test set. The ML algorithm was developed using the Python 3.9 (Wilmington, DE, USA) programming language with Scikit-learn, a free software ML library [35].

## 3. Results

### 3.1. Plasma Values of miRNAs

miRNAs were collected from the plasma of 78 healthy donors, randomly selected, and grouped into four age classes: young adults (22–50 years, *n* = 19), adults (51–70 years, *n* = 28), older adults (71–99 years, *n* = 20), and ultra-centenarians (100–111 years, *n* = 11).

Figure 1 shows the distribution of miRNAs levels in relation to age. For miR-21-5p and miR-126-3p, it is apparent that the maximum values were found in the age ranges 51–70 and 71–99 years, while the ultra-centenarians showed lower levels and the smallest variability. For the miR-146a-5p and miR-181-5p, in all classes of age, it was observed that the median value is around 5 but ultra-centenarians show a higher variability than the other classes of age. Individual points are considered as outliers if, when defining the upper adjacent value U = q_3_ + (3/2) (q_3_ − q_1_) and the lower adjacent value L = q_1_ − (3/2) (q_3_ − q_1_), the point is higher than U or lower than L. The upper (U) and lower (L) adjacent values are defined by Tukey (1977).

Regarding significance, the Kruskal–Wallis test clearly demonstrates that all the four miRNAs show significant age-related variability (Table 1).

All data (mean, median and standard deviation of the total samples and of the samples divided by gender) and the *p*-values of the comparisons between the different age groups are presented in Table 1. Given the relatively small number of the various groups, there was no significance found between groups of age by sex.

The Table 1 shows mean, median, and standard deviation (SD) of the four miRNAs in different population age groups and for gender. The letters a, b, c, and d indicate, respectively, young adults (22–49), adults (50–69), older adults (70–98), and ultra-centenarians (100–111). The Kruskal–Wallis test and Dunn test with Bonferroni adjustment are used to test significant differences among age groups for each miRNA.

Data show that for the miR-21-5p values there was no significant difference between the ultra-centenarian and young people (data not shown) while the ultra-centenarian values were lower and significantly different from those of adults and older adults. For miR-126-3p, the values for ultra-centenarians were significantly lower than those of other groups that were not significantly different from each other. For the miR-146a-5p and miR-181-5p values, there was no significant difference between ultra-centenarian values and the other group due to the high heterogeneity of the centenarian measures. The plasma levels of miRNA of the young adults were lower than those of the other adult groups but the significance with both groups was obtained only for miR-181-5p.

### 3.2. Correlation of Plasma Values of miRNAs with Some Parameters

Robust multiple regression models were considered to estimate the relationship between each miRNA with some individual and lifestyle-variables, i.e., gender, age, BMI, and smoking habits.

The estimation method was used to correct the standard errors of estimated coefficients in the presence of heteroskedasticity. Results of the estimated coefficients are reported in Table 2 and Table 3. Figure 2 reports the scatterplots of the estimated models and the observed data for each miRNA by age.

As the relationship between miR-21-5p and age seems to follow a quadratic function, a quadratic model for age was estimated:(1)$$miR_{-21-5p}=b_{0}+b_{1}Age+b_{2}Age^{2}+b_{3}BMI+b_{4}Smoke+b_{5}Gender$$

The miR-21-5p increases with age, with a decreasing average rate of change. A higher level of miR-21-5p was observed around the age 60–65 years (Figure 1. at the top left) and after these ages it tended to shrink to the same levels as in younger participants. No significant effects, at a 5% significance level, were found for gender, BMI, and smoking habits.

For the other miRNAs, linear relationships with age were estimated as follows:$$miR{-}_{126-3p}=b_{0}+b_{1}AgeClass+b_{2}BMI+b_{3}Smoke+b_{4}Gender$$
$$miR{-}_{146-5p}=b_{0}+b_{1}AgeClass+b_{2}BMI+b_{3}Smoke+b_{4}Gender$$
(2)$$miR{-}_{181a-5p}=b_{0}+b_{1}AgeClass+b_{2}BMI+b_{3}Smoke+b_{4}Gender$$

The estimated model for the miR-126-3p shows that in the centenarians compared to young cells, the level of miR-126-3p reduced by a mean of −3.63. The category of ho has never smoked vs. who smoked show an increasing mean level of 2.91. No significant effect exists for gender and BMI. For the miR-146a-5p, no significant effect was noted for all the analysed variables. The levels of the miR-181a-5p increased in ex-smokers and in never smokers compared with smokers and no difference exists for gender in all miRNAs.

### 3.3. miRNAs Levels in HUVECs Undergoing Replicative Senescence

Since the ECs senescence is known to be involved in the ageing process, we hypothesised that the observed age-related alterations of the miRNA levels in study could be linked to EC senescence. Because it is difficult to study EC senescence ex vivo, to investigate age-related changes of miRNA, HUVECs were used as an in vitro model. In cultured HUVECs undergoing replicative senescence, miRNA levels were measured in CM. Cell senescence was defined based on cumulative population doubling (CPD). We considered CPD > 56 for old (senescent) cells, CPD < 32 for intermediate age cells, and CPD < 20 for young cells. Senescence was estimated by SA-β-gal activity and measured as the percentage of positive cells (40 ± 10 in old, 10 ± 4 in intermediate age and 2 ± 1 in young cells, data not shown). As shown in Figure 3, the CM of old ECs showed higher amounts of the four miRNAs than for young cells. The same results were obtained with the CM of intermediate age ECs for two out of four of the studied miRNAs (miR-21-5p and miR-126-3p) that were present in higher amounts than in the CM of young cells. In addition, the CM of old ECs showed higher amounts of miR-146a-5p and miR-181a-5p than CM of intermediate cells. These results indicate that in vitro EC miRNA levels correlate with the replicative senescent state and suggest that variations observed ex vivo in adult and older adults might be linked to the EC senescence process (Figure 3).

### 3.4. Enriched KEGG Pathway Clustered by Validated Targets of miR-21-5p, miR-126-3p, miR-146a-5p, and miR-181a-5p and Corresponding Target Genes

In order to investigate the possible regulation mechanisms involved in the ageing process of the four miRNAs under study we utilized an online bioinformatics database, called miRWalk, to select plausible targets and validated targets of these miRNAs. Considering that miR-21-5p and miR-126-3p display similar patterns while miR-146a-5p and miR-181a-5p show a linear trend, we analyzed the two pairs of miRNAs separately. The potential target genes of miR-21-5p and miR-126-3p were 448 and 271, respectively. A total of 27 genes were recognized as common targets of both miR-21-5p and miR-126-3p by overlapping the analyses. Results of the enrichment KEGG pathway indicated that the targeted genes regulated by miR-21-5p and miR-126-3p were involved in 105 pathways of which the top 15 are shown in Appendix A. Interestingly, among the most significant pathways, two are related to longevity regulating pathways. In total, 1492 and 1475 target genes of miR-146a-5p and miR-181a-5p were, respectively, predicted and 52 genes were co-regulated by miR-146a-5p and miR-181a-5p. The target genes of miR-146a-5p and miR-181a-5p were significantly enriched in 207 KEGG pathways. Notably, among the first 15 pathways, three are related to the Sumoylation process, an essential post-translational modification that has evolved to regulate intricate networks within emerging complexities of eukaryotic cells [36]. See Appendix A for the complete analyses.

### 3.5. ML Analyses

The results of *ML* are represented in Figure 4. For calculation, miRNAs, C reactive protein, telomeres, paraoxonase (PON), trolox equivalent antioxidant capacity, and malondialdehyde values were considered according to age-ranges. Overall, 100% of subjects under 50 years old have a telomere length of above 0.903, an miR-181a-5p of below 4.606, a miR-21-5p of below 17.04, and an miR-126 -3p of less than 13.586. Individuals with a telomere length greater than 0.903 and a miR-181a-5p greater than 4.606 were between 50 and 70 years. Individuals with a telomere length below 0.903, a miR-21-5p greater than 5.164, a miR-181a-5p less than 11.441 and a PON less than 140.813 were between 71 and 99 years. For ultra-centenarians, the telomere length is always less than 0.903. Additionally, they have an miR-21-5p below 5.164. We also found ultra-centenarians with an miR-21-5p higher than 5.164 but all of them would have had an miR-181a-5p above 11.441.

## 4. Discussion

Ageing is a series of physiological events that are usually ineluctable. It becomes a risk and accelerator factor for most age-related diseases, including cardiovascular ones [37]. In turn, senescence is the biological ageing of cells and an irreversible form of long-term cell-cycle arrest, caused by excessive intracellular or extracellular stress or damage. ECs senescence impairs vascular functions, thereby enhancing the ageing of tissues and organs. Several stimuli, including reactive oxygen species, inflammatory cytokines, and telomere dysfunction, can increase their senescence [17]. Recently, senescent ECs have emerged as a possible source of circulating miRNAs that are short non-coding RNAs that generally either induce the degradation of mRNA or repress the translation of target transcripts. It has already been established that they are important regulators of the ageing process and modulators of longevity. Different miRNAs directly influence the duration of life through the modulation of various ageing pathways which represent adaptive mechanisms aimed at maintaining the homeostasis of the organism including inflammatory responses [4].

In the present study, we analysed the blood levels of four circulating miRNAs, namely miR-146a-5p, miR-126-3p, miR-21-5p, and miR-181a-5p, which are involved in several pathways related to inflammation and EC senescence, in 78 healthy subjects aged between 22 to 111 years.

Most studies on centenarians describe them as the best models of successful ageing, representing selected people in which the appearance of major age-related diseases, such as cancer and cardiovascular diseases, has been consistently delayed or escaped. However, extreme longevity is often characterized by a non-unique and unambiguous phenotype, as demonstrated by the centenarian population in the world. Not all centenarians are similar, although all can represent a model of “positive biology” [38,39].

The data reported that for miR-126-3p and miR-21-5p, the maximum values were found between 51 and 99 years while the ultra-centenarians showed lower levels. For miR-146a-5p and miR-181a-5p, the highest mean values were observed in the group of individuals over 100 years old, but the differences with the values of younger people were not significant due to their heterogeneity as previously stated.

The Kruskal–Wallis test showed that all four miRNAs had significant age-related variability. Then, robust multiple regressions were performed to estimate the relationship between each miRNA with some parameters in addition to age, i.e., gender, BMI, and smoking habits.

No effect of gender and BMI was observed in relation to all circulating miRNAs levels, unlike the results of Ameling et al., who found a significant association of some miRNAs with BMI, likely due to a different composition of the study population with a wider range of BMI values [40].

Never-smoking is instead related to an incremental effect of miR-126-3p and miR-181-a-5p. This datum is not surprising because several miRNAs have been shown to be under regulated in smokers [41].

Finally, a significant effect with age was observed for miR-126a-3p and miR-21-5p. For the levels of miR-181-a-5p no significant age effect was observed.

In previous reports in other samples of Sicilian populations, results have demonstrated that in addition to miR-181a-5p the blood levels of miR-223-5p and let-7a-5p also did not show significant differences between centenarians and adult females (18–64 years old) [42]. Unfortunately, in that paper, older women (65–99 years old) were not included. In another report, mean levels of miR-126-3p, carried in small extracellular vesicles (EV) isolated from plasma, increased significantly with age, with the highest levels observed in the nineties [2].

Thus, we assumed that the age-related miRNAs variations in adult and older adults could be linked to ECs senescence. Based on the evidence that in vitro replicative senescence of ECs mimics the progressive age-related changes of endothelial functions described ex vivo, we used HUVECs undergoing replicative senescence as an in vitro model [43]. We observed that HUVECs released miRNA at progressively higher levels with increasing senescence processes except for miR-146a-5p and miR-181a-5p, whose intermediate values were not significantly different from those observed in younger cells. In a previous paper, HUVECs also released EVs-miR-126-3p in progressively higher levels with increasing senescence process. Overall, these data suggest that variations observed ex vivo from young to older people might be related to the EC senescence process [38].

Finally, results from ML highlighted the significant role, in people over 70 year, played by miR-21-5p and miR-181a-5p, in addition to PON and telomere length [44,45].

Circulatory miRNAs change with age and with the development of age-related diseases. This suggests their possible use as non-invasive “biomarkers of ageing”, which can predict successful ageing and longevity [8].

Some other studies have investigated the differential expression of miRNA between young, old and centenarian populations. Balzano et al. performed a study comparing circulating miRNA levels in three cohorts, i.e., centenarians, patients with rheumatoid arthritis in treatment with corticosteroids, and young and middle-aged healthy subjects as controls [13]. Their results showed that miR-425-5p, miR-21, and miR-212 were significantly decreased in centenarians and in patients with rheumatoid arthritis treated with corticosteroids compared to controls. The authors suggest that this miRNAs pattern could protect against inflammation for centenarians and also found that this occurs in patients treated with glucocorticoids. Serna et al. studied the miRNA expression profiles in mononuclear cells of centenarians, octogenarians, and younger Spaniards [14] The results showed a significant overlap of the profiles of centenarians with those of the young but not with the octogenarians. A longitudinal study was performed by Smith-Vikos et al. using serum samples collected from short-lived (58–75 years) and long-lived (76–92 years) participants of the Baltimore Longitudinal Study of Ageing [15]. A total of 24 miRNAs were found to be significantly upregulated and 73 were found to be significantly downregulated in long-lived individuals. Six of these miRNAs, not analysed in the present study, were both differentially expressed and correlated with individual lifespan.

Recent data on inflammation-related miRNAs in successful and unsuccessful ageing outline a complex scenario characterised by an altered expression of specific miRNAs which mainly target the nuclear factor κB (NF-κB) pathway, the master regulator of inflammation. In physiological conditions, their transcription is at baseline levels. However, the initiation of pro-inflammatory signalling results in the strong co-induction of their expression through a mechanism that is largely NF-κB-dependent [46]. Furthermore, miRNAs have been reported to show age-dependent changes in blood levels, and to be markers of inflamm-ageing, i.e., the low-grade inflammatory status of ageing, in successful and unsuccessful ageing [11,12]. miR-146a-5p was the first to be identified as an NF-κB-dependent miRNA, being up regulated in response to various immune stimuli. In turn, miR-146a downregulates relevant proteins which are also in the canonical NF-κB pathway. miR-146a exerts several beneficial effects on successful ageing, via a fine tuning of canonical and non-canonical NF-κB pathways [23,47]. A decline in miR-146a-5p levels would be expected to trigger the release of inflammatory cytokines [23]. Accordingly, the downregulation of miR-146a has been observed in patients with diabetes, obesity, and hypertension [48]. An increased expression of miR-146a in HUVECs and in aortic and coronary ECs during replicative senescence occurs, thus demonstrating that NF-κB activation and cell senescence can be modulated by the miRNAs described. Thus, cellular senescence and inflammation signalling activation may be closely interconnected and share common regulators [11].

Furthermore, miR-21-5p, induced by several pro-inflammatory molecules, is able to activate NF-κB and Nod Like Receptor family pyrin domain containing 3 (NLRP3), hence orchestrating the fine tuning of the inflammatory response through direct and indirect activities on NF-κB and NLRP3 pathways in a context-dependent manner. In vitro and in vivo studies of animal models confirmed the essential role played by miR-21 in regulating this inflammatory switch, both promoting or inhibiting NF-κB/NLRP3 pathways. For all these reasons, miR-21 can be considered as an miRNA that is able to modulate the “switch on/off” of inflammation at appropriate times [11]. Accordingly, miR-21-5p is down-regulated in healthy older people and up-regulated in patients affected by many age-associated disorders, representing an excellent candidate with which to track the ageing trajectories. A significant miR-21-5p increase observed in Alzheimer’s patients with severe cognitive impairment supports this hypothesis [49]. miR-21 also lies at the intersection of senescence, inflammation, and age-related diseases [12].

It has been demonstrated that the age-related increase in plasma miR-126-3p was paralleled by a 5/6-fold increase in intra/extracellular miR-126-3p in cultured HUVECs undergoing senescence. Indeed, it has a role in the maintenance of endothelial integrity, enhancing endothelial functions and promoting blood vessel formation. Senescence-associated miR-126-3p up-regulation is likely a senescence-associated compensatory mechanism. It is also a modulator of inflammation and the innate immune response, targeting some components of the NF-kB pathway [50].

Finally, a target prediction and pathway analysis of miR-181a-5p showed overlapping involvement of the inflammatory pathways, consistent with the relationship between chronic inflammation and ageing [51]. The upregulation of miR-181a may be associated with the homeostatic response to inflammatory stimuli [52]. In a previous paper, we presented a case report of a centenarian woman in relatively good health, despite laboratory signs of atrophic gastritis and a chronic status of inflammation [42]. Her levels of miR-181a-5p, miR-223-5p, and let-7a-5p, which play a role in the control of innate immunity and inflammation, were higher than those of female centenarians. These data suggested a possible epigenetic modulation, likely with anti-inflammatory effects that confer protection against tissue damage. A decreased expression of miR-181a has been observed in patients with coronary artery disease and it has been suggested to have an antiatherogenic effect through blocking NF-κB activation and vascular inflammation [53].

Furthermore, the target pathway analysis of miR-126-3p and miR-21-5p also demonstrated their involvement in the nutrient sensing pathway that is linked to insulin and insulin growth factor-1, which has an important role as “gatekeeper” by balancing the cell response to oxidative stress and nutrient availability. Downstream of this pathway there is the Forkhead box O3 (FOXO3) A transcription factor. Notably, the FOXO family transactivate genes are involved in resistance to oxidative stress, energy metabolism, DNA damage repair, glucose metabolism, autophagy and the protection of proteins by chaperones, favouring survival and longevity [54]. Instead, a target pathway analysis of miR-146a-5p and miR-181a-5p also illustrates their involvement in the sumoylation process. Sumoylation plays critical roles in cellular senescence. Enhancing the global sumoylation or inhibiting the desumoylation process seems to promote senescence. In sumoylation-mediated cellular senescence, the p53 and RB proteins are SUMO substrates and have been identified as important molecules in this senescence process [55,56].

Overall, the data suggest that the analyzed miRNAs can be active components of the senescent ECs secretome and may modulate the rate of inflammation at the cellular and systemic level [12]. Tissue and circulating miRNAs could contribute to restraining the activity of the senescent cell secretome and modulate the destruction induced by the activation of the inflammatory response [12,42,50].

With age, cellular senescence, visceral obesity, microbiota, and the stimulation of the immune system contribute to inducing and perpetuating inflamm-ageing over time, inducing an upregulation of these miRNAs, also called inflamma-miRs, that lead to the excessive activation of the inflammatory pathways [10,11,12]. Thus, the up-regulation of inflamma-miRs in blood circulation occurs in healthy older individuals. However, as our data show, this increase is not significant (although with a great heterogeneity of response, see above) in ultra-centenarians since they are able to control inflammation. On the other hand, their values are greater in patients with age-related diseases. So, circulating levels could be useful biomarkers of these diseases and of their complications. The main sources of inflamma-miR circulating in successful and unsuccessful ageing should be both immune cells and ECs [9,11,23,57]. Accordingly, our data show that variation observed ex vivo from younger to older people might be related to the ECs senescence process.

## 5. Conclusions

Our data provides further evidence of these four miRNAs acting as biomarkers of successful and unsuccessful ageing, although a note of caution should be added considering the small sample size and the imbalance of the number of samples for each age group. We confirmed that miR-126-3p and miR-21-5p are correlated with ageing and it is therefore possible to consider them hallmarks of bio-positive ageing. Moreover, ML data for miR-21-5p and miR-181a-5p suggest that these miRNAs can be ideal indicators of longevity along with telomere length. Indeed, correlation analyses show that miR-181a-5p is lower in never-smoked subjects. Thus, circulating inflamma-miRNAs might have diagnostic/prognostic relevance for age-related diseases.

## Figures and Tables

**Figure 1 cells-11-01505-f001:**
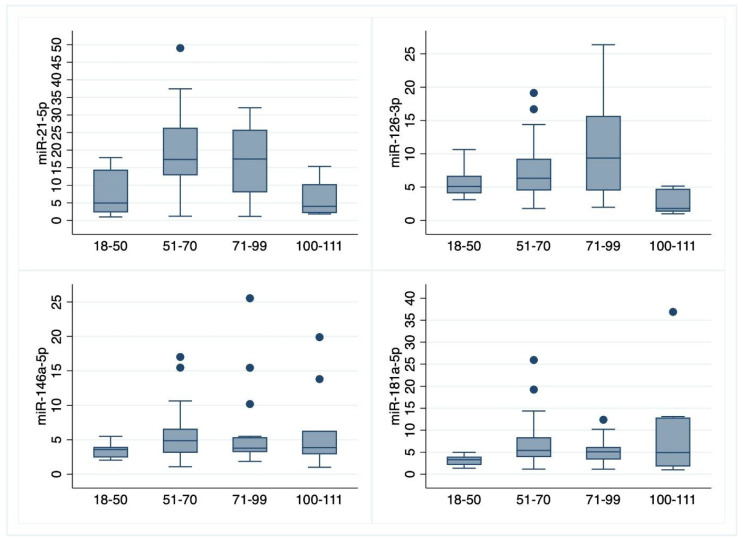
Box-and-Whisker plots of miRNAs levels by class of age. Box-and-Whisker plots report minimum, quartiles (q_1_, q_2_ and q_3_) and maximum levels of each miRNA by age class. The individual points (or dots) plotted are outliers. The spacings between different parts of the box indicate dispersion and skewness in the data and shows a graphical measure of interquartile range (q_3_–q_1_) and range (max–min).

**Figure 2 cells-11-01505-f002:**
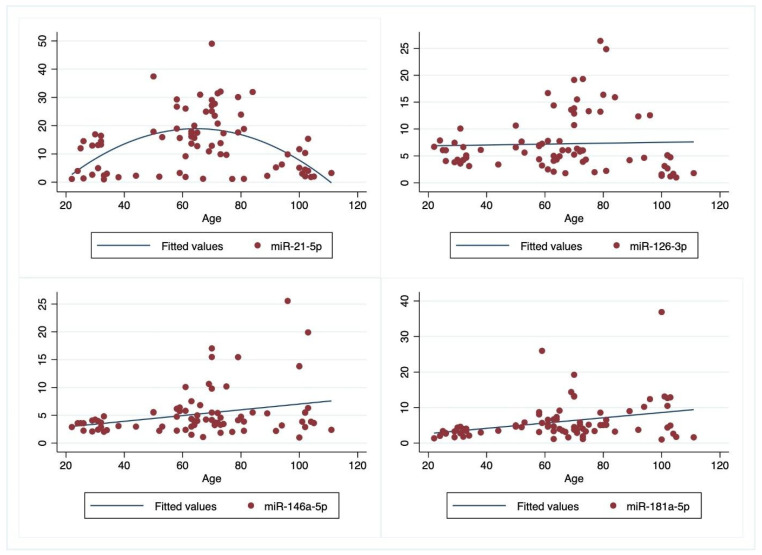
Scatter Plot of observed and fitted models of correlation of plasma values for miRNAs with age.

**Figure 3 cells-11-01505-f003:**
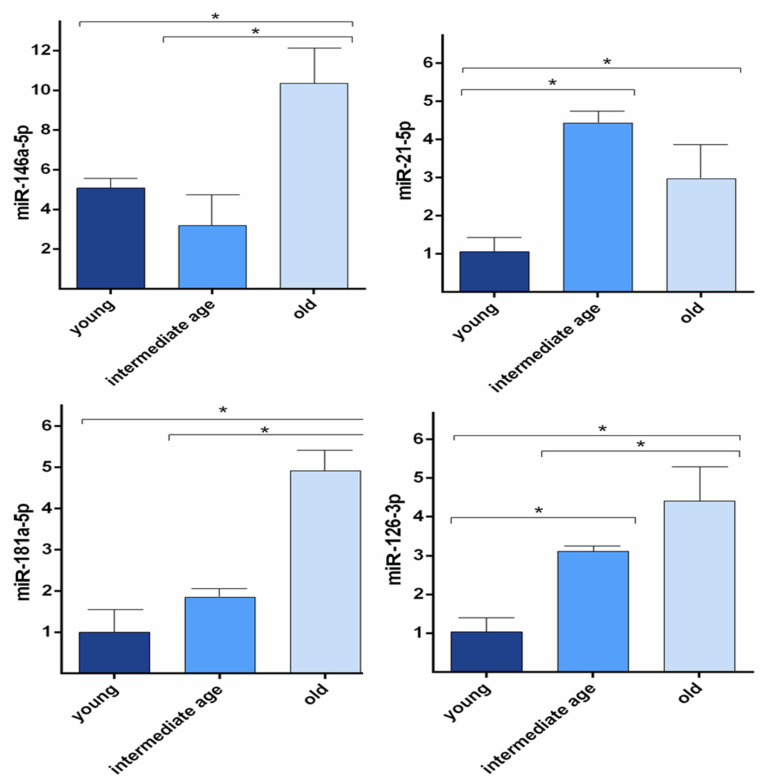
miRNAs levels variation in HUVECs undergoing senescence. Bar charts show extracellular miR-146a-5p, miR-21-5p, miR-181a-5p, miR-126-3p levels in young, intermediate, and old (senescent) HUVECs. Data were calculated by qRT-PCR and represent mean ± SD of three different experiments analysed. CTs (cycle thresholds) resulting from qRT-PCR analysis were normalised with miR-30a; levels were calculated with 2-DCT method and expressed as folds, with respect to lowest value registered. Comparisons among multiple groups were analysed by one-way analysis of variance, followed by Bonferroni’s post hoc test. * *p* < 0.05.

**Figure 4 cells-11-01505-f004:**
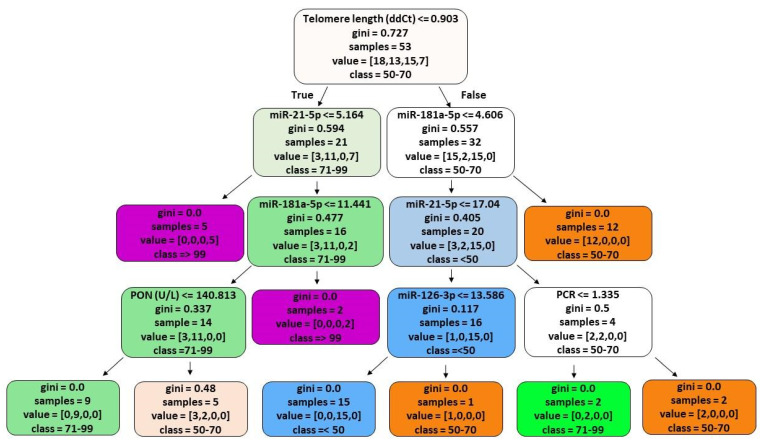
Figure shows the machine learning results. For the interpretation of the decision tree, it is necessary to know that each node of a decision tree contains a condition. This condition can be true or false. If the condition is true, we descend to the next left node. If the condition is false, we descend to the next right node. The different colors represent the age groups (light blue < 50, orange 50–70, green 71–99, pink/lilac > 99 years).

**Table 1 cells-11-01505-t001:** Plasma miRNA levels in the 4 age-groups of Sicilian population. Please, note that a, b, c, and d are the letters used to indicate the different groups: a = Young Adults; b = Adults; c = Older Adults; d = Ultracentenarians.

**miRNA**		**(a) Young Adults** **(22–50 y.o.)** **N = 19, M = 7, W = 12**			**(b) Adults** **(51–70 y.o.)** **N = 28, M = 14, W = 14**			**(c) Older Adults** **(71–99 y.o.)** **N = 20, M = 13, W = 7**			**(d) Ultracentenarians** **(100–111 y.o.)** **N = 11, M = 2, W = 9**			***p*-Value of Kruskal–Wallis Test**	***p*-Value Bonferroni Test**
		mean	median	SD	mean	median	SD	mean	median	SD	mean	median	SD	*p* = 0.0003	(a vs. b), *p* = 0.0028; (a vs. c), *p* = 0.0284; (b vs. d), *p* = 0.0024; (c vs. d), *p* = 0.0171.
**miR-21-5p**	**N**	**8.22**	**4.96**	**6.45**	**18.87**	**17.34**	**11.06**	**16.73**	**17.47**	**10.74**	**5.73**	**4.01**	**4.58**		
	M	7.71	4.96	6.35	19.56	18.07	13.24	15.40	17.32	8.75	3.05	3.05	1.36		
	W	8.51	7.50	6.78	18.18	17.34	8.83	19.19	27.77	14.19	6.33	4.43	4.88		
**miR-126-3p**	**N**	**5.70**	**5.10**	**2.16**	**7.58**	**6.33**	**4.54**	**10.77**	**9.36**	**7.33**	**2.67**	**1.80**	**1.62**	*p* = 0.0002	(a vs. d), *p* = 0.0238; (b vs. d), *p* = 0.0005; (c vs. d), *p* = 0.0001.
	M	6.27	6.06	2.19	6.98	5.84	4.80	11.85	12.36	8.21	3.72	3.72	1.46		
	W	5.37	4.49	2.17	8.18	6.71	4.35	8.76	6.35	5.30	2.44	1.68	1.64		
**miR-146a-5p**	**N**	**3.34**	**3.57**	**0.94**	**3.34**	**3.57**	**0.94**	**5.67**	**3.79**	**5.63**	**6.23**	**3.85**	**5.63**	*p* = 0.0491	(a vs. b), *p* = 0.0177.
	M	3.11	3.57	0.78	5.41	3.71	4.92	5.05	3.43	6.23	4.60	4.60	2.43		
	W	3.48	3.32	1.03	6.13	5.66	2.47	6.83	5.41	4.52	6.60	3.85	6.17		
**miR-181a-5p**	**N**	**3.11**	**3.28**	**1.03**	**7.22**	**5.41**	**5.52**	**5.27**	**5.07**	**2.85**	**9.30**	**4.93**	**10.36**	*p* = 0.0024	(a vs. b), *p* = 0.0006; (a vs. c), *p* = 0.0358.
	M	2.87	2.82	0.78	6.30	4.83	4.73	4.99	4.03	3.12	4.65	4.65	0.40		
	W	3.25	3.43	1.16	8.15	6.62	6.24	5.80	5.19	2.39	10.34	10.47	11.30		

**Table 2 cells-11-01505-t002:** Coefficients and significance estimates of miR-21-5p.

R^2^ = 0.346	Coefficient	*p*
**Age**	1.46	<0.0005
Age^2^	−0.01	<0.0005
**BMI**	−0.20	=0.54
**Smoke**		
Smoker (reference)		
Ex-smokers	1.94	=0.57
Never smoked	5.24	=0.06
**Gender**		
M (reference)		
F	−2.02	=0.40
b_0_ (constant)	−23.32	=0.01

**Table 3 cells-11-01505-t003:** Coefficients and significance estimates of other miRNAs.

	miR-126-3p		miR-146a-5p		miR-181a-5p	
	Coefficient	*p*	Coefficient	*p*	Coefficient	*p*
**Age Class**						
22–50 (Reference)						
51–70	2.08	=0.14	2.84	<0.0005	4.23	0.00
71–99	4.37	=0.05	2.39	=0.16	1.61	0.30
100–111	−3.63	<0.0005	1.15	=0.34	5.28	0.13
**BMI**	0.07	=0.61	−0.05	=0.74	−0.01	0.95
**Smoke**						
Smoker (Reference)						
Ex-smoker	2.05	=0.25	2.14	=0.10	4.13	0.00
Never smoked	2.91	=0.04	1.36	=0.06	1.93	0.01
**Gender**						
M (Reference)						
F	−0.85	=0.49	0.72	=0.46	1.63	0.16
*b* _0_	2.50	=0.49	2.95	=0.44	0.68	0.87
	R^2^ = 0.2779		R^2^ = 0.116		R^2^ = 0.2169	

Our elaboration was conducted with Stata Software 16.1.

## Data Availability

The data presented in this study are available in [insert article or Appendix A here].

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
