# Peer review of "miR-126-3p and miR-21-5p as Hallmarks of Bio-Positive Ageing; Correlation Analysis and Machine Learning Prediction in Young to Ultra-Centenarian Sicilian Population"

_cells, 2022, doi:10.3390/cells11091505_

Round 1

Reviewer 1 Report

This manuscript is practical and innovative to explore the extracellular circulating miRNAs in pathways related to inflammation and ECs senescence. This study shed new light on diagnosing successful and unsuccessful aging.

There are some shortcomings in this paper . Such as the authors claimed that “Indeed, the secretion of specific miRNAs seem to be tissue- and age-associated. They have been found to be secreted into microvesicles and exosomes or free in plasma, in systemic and local environments, where the action of ribonucleases is limited” However, there is no concrete data or references to explain the secretion of the miRNAs into micro vesicles, exosomes or plasma. Please provide more details.

In Introduction part, your own contributions can be simplified using less words. The details of the experiment design can be introduced in the Materials and Methods.

Subsection 2.3: this subsection concerning quantitative PCR (qPCR) is highly insufficient being very brief and lacking of vital information. For instance, there are no details regarding assay efficiencies, primer sequences and concentrations. I'd highly recommend adherence to the 'MIQE-guidelines' (Bustin et al., 2009). 

According to the line 143,the authors used miR-30a for the normalisation of RT-qPCR data, using the 2-DCT method. They should provide the reason why miR-30a was chosen for normalization. Recently, multiple internal reference were applied in miRNA studies, such as miR-16a, RNU6. It is of great importance to explain the reasons for using miR-30a for normalization. I also suggested that the authors provide convincing evidence ( which may require more experimental results) to prove that miR-30a is indeed suited.

Subsection 2.1: The population was divided in 4 age groups, i.e., young adults (22–50 117years, n=19), adults (51–70 years, n=28), older adults (71–99 years, n=20), Please provide the reason or reference why the the grouping was designed in this way.

It is suggested to provide the target genes and related signal pathways of the miRNA involved in this paper, and these contents can be reflected in the experimental results.

This manuscript need to edit carefully in much more concise. There are two ways in the naming of two miRNAs. Such as “For the mir-146a-5p” in line 244 and “mir181-5p” in line 248

Table 2b is too big. Please adjust the words or graphics in a suitable form.

Author Response

REVIEWER 1

Comments and Suggestions for Authors

This manuscript is practical and innovative to explore the extracellular circulating miRNAs in pathways related to inflammation and ECs senescence. This study shed new light on diagnosing successful and unsuccessful aging. There are some shortcomings in this paper.

Q1. Such as the authors claimed that “Indeed, the secretion of specific miRNAs seem to be tissue- and age-associated. They have been found to be secreted into microvesicles and exosomes or free in plasma, in systemic and local environments, where the action of ribonucleases is limited”. However, there is no concrete data or references to explain the secretion of the miRNAs into micro vesicles, exosomes or plasma. Please provide more details.

A1. We thank the reviewer for the possibility to improve our manuscript, so we added new references and we completely modified the introduction. Please see the text.  

Q2. In the Introduction part, your own contributions can be simplified using less words. The details of the experiment design can be introduced in the Materials and Methods.

A2. We modified the Introduction part and, as suggested by the Reviewer 3, we rewrited completely the text of this section removing the details of the experiment design. Please see the text.

Q3. Subsection 2.3: this subsection concerning quantitative PCR (qPCR) is highly insufficient being very brief and lacking of vital information. For instance, there are no details regarding assay efficiencies, primer sequences and concentrations. I'd highly recommend adherence to the 'MIQE-guidelines' (Bustin et al., 2009). 

A3. The expression levels of miRNAs were evaluated by miScript Primer Assay, specific for each microRNAs (purchased from Qiagen). These PCR assays are provided in ready-to-use mode by Qiagen and every assay is a "miScript miRNA PCR Array" that has been bench validated to ensure sensitive and specific detection of a mature miRNA via real-time PCR. Unfortunately, Qiagen does not provide details regarding assay efficiencies such as primer sequences.

Q4. According to the line 143, the authors used miR-30a for the normalisation of RT-qPCR data, using the 2-DCT method. They should provide the reason why miR-30a was chosen for normalization. Recently, multiple internal reference were applied in miRNA studies, such as miR-16a, RNU6. It is of great importance to explain the reasons for using miR-30a for normalization. I also suggested that the authors provide convincing evidence (which may require more experimental results) to prove that miR-30a is indeed suited.

A4. The choice of the appropriate reference for circulating microRNAs normalization remains problematic, there is no unanimous consensus on the best reference to use. The use of RNU6 has been questioned by Schwarzenbach, Heidi, et al. "Data normalization strategies for microRNA quantification." Clinical chemistry 61.11 (2015): 1333-1342.  We used mir-30a for normalization based on the following papers: “Marabita F, De Candia P, Torri A, Tegnér J, Abrignani S, Rossi RL. Normalization of circulating microRNA expression data obtained by quantitative real-time RT-PCR. Brief Bioinform. 2016; 17:204–12. https://doi.org/10.1093/ bib/bbv056.” “Yuan T, Huang X, Woodcock M, Du M, Dittmar R, Wang Y, Tsai S, Kohli M, Boardman L, Patel T, Wang L. Plasma extracellular RNA profiles in healthy and cancer patients. Sci Rep. 2016; 6:19413. https://doi.org/10.1038/srep19413”.

Q5. Subsection 2.1: The population was divided in 4 age groups, i.e., young adults (22–50 117years, n=19), adults (51–70 years, n=28), older adults (71–99 years, n=20), Please provide the reason or reference why the the grouping was designed in this way.

A5. We thank you for this observation. We utilized the same age-groups gradient using in the paper: “Can Be miR-126-3p a Biomarker of Premature Aging? An Ex Vivo and In Vitro Study in Fabry Disease. Cells 2021, 10, 356. doi: 10.3390/cells10020356.” to demonstrate the correlation between miRNAs and age-related diseases.  

Q6. It is suggested to provide the target genes and related signal pathways of the miRNA involved in this paper, and these contents can be reflected in the experimental results.

A6. Thanks to the reviewer for the suggestion, we added a new paragraph (3.4) on pathways and genes targeted by the miRNAs selected in this manuscript. Please see the text and the Supplemental materials.

Q7. This manuscript need to edit carefully in much more concise. There are two ways in the naming of two miRNAs. Such as “For the mir-146a-5p” in line 244 and “mir181-5p” in line 248.

A7. Thanks to the reviewer for the possibility to improve our manuscript. We edited the manuscript and corrected miRNA names.

Q8. Table 2b is too big. Please adjust the words or graphics in a suitable form.

A8. Done.

Reviewer 2 Report

This paper by Accardi et al. investigated the interconnection between the level of four extracellular miRNAs in human endothelial cells and age. The study added to the knowledge on the identification of miRNAs as biomarkers of human aging. I think the topic of this work is of good importance and interest, but there are several points below needed to be further addressed before I can recommend its publication to Cells.

Major comments:

  1. What are the reasons to specifically pick these four miRNAs, have the authors looked at other miRNAs, and what might be the correlation look like?
  2. The last paragraph of the introduction section can be clearly improved. As part of the introduction, it would be necessary to explicitly clarify what is the input of the ML model, what is the output, what is the motivation for using the ML model, what is the main conclusion from using the ML model, etc.
  3. The conclusion of the manuscript might be affected by the fact of the small sample size (78) and the imbalance of the number of samples in each age group, e.g. number of samples in adults (28) is much higher than that in ultra-centenarians (11). The authors need to provide more discussion on this point.
  4. What are the features in predicting the age of the machine learning (ML) model? Did the authors do feature selection? Is the ML model (decision tree) trained on only 78 samples? Would that be a problem in having a very high possibility of overfitting?
  5. Did the authors try other methods of supervised ML model, e.g. from the simplest linear regression to kernel regression, with and without regularization to compare with the decision tree model used in the manuscript, and see which model would perform the best?

Minor comments:

  1. The authors need to check the texting in the current PDF. For example, in line 219, mojibake was found.
  2. The quality of the figure, e.g. Figure 4, can clearly be improved.

Author Response

REVIEWER 2

Comments and Suggestions for Authors

This paper by Accardi et al. investigated the interconnection between the level of four extracellular miRNAs in human endothelial cells and age. The study added to the knowledge on the identification of miRNAs as biomarkers of human aging. I think the topic of this work is of good importance and interest, but there are several points below needed to be further addressed before I can recommend its publication to Cells.

Major comments:

Q1. What are the reasons to specifically pick these four miRNAs, have the authors looked at other miRNAs, and what might be the correlation look like?

A1. Thanks to the reviewer for the possibility to improve our manuscript, as also suggested by the reviewer 1 and 3, we modified the introduction and we better explained why we focused our attention on selected miRNAs. Moreover, we added rationale and references, in order to guide the readers in understanding the results, and a new paragraph on gene target analysis (see paragraph 3.4. Enriched KEGG pathway clustered by validated targets of miR-21-5p, miR-126-3p, miR-146a-5p, and miR-181a-5p and corresponding target genes; see also Supplemental materials Tables 2-5 and Figures 1,2).

Q2. The last paragraph of the introduction section can be clearly improved. As part of the introduction, it would be necessary to explicitly clarify what is the input of the ML model, what is the output, what is the motivation for using the ML model, what is the main conclusion from using the ML model, etc.

A2. We thank the reviewer for this comment. The paragraph has been rewritten to clarify several issues related to the use of a ML technique for classifying subjects in several ranges of ages, and finding which features are relevant in the clustering process. We added the following paragraph:

“Furthermore, since some specific biological features seem to be age-associated, it was possible to find the relation between these features and people's age, using machine learning (ML) techniques. ML is a branch of Artificial Intelligence focused on imitating the way human beings are capable of learning, taking experimental data and examples as inputs to infer a specific result or output (25). ML often builds mathematical models based on sample data to make predictions without being explicitly programmed for such purposes. In general, ML models are considered as black boxes where inputs are transformed into outputs by combining such inputs with a set of adjusted data obtained as a result of a training process (26). Although ML provides many benefits in different areas of application, the lack of interpretability could be a problem in some circumstances. In the context of this work, the age prediction problem by considering several biological features is also addressed, since it is interesting to study which factors affect the population from young to ultra-centenarians. Finding such relations in an interpretable manner is important, therefore, it is necessary to use ML techniques with a high degree of interpretability. Consequently, a decision tree classifier has been selected for this purpose, since such ML method presents a high rate of accuracy, great robustness and results are very interpretable as long as they are short.”

Please, see as well that:

  1. A) In section 3.5. of the Results, inputs are defined: “(…) For calculation, it was considered miRNAs, C reactive protein, telomeres, paraoxonase (PON), trolox equivalent antioxidant capacity, and malondialdehyde values according to age-ranges”.
  2. B) And the main conclusion for using the model is addressed in the following paragraph found in Section 3, Discussion: “Finally, results from ML pointed out a significant role, in people over 70 years, played by miR-21-5p and miR-181a-5p, in addition to PON and telomere length (46,47).

As it is observed, this ML method is very interpretable and it is possible to get conclusions about which features are significant during the process of classification. This degree of interpretability is not obtained with other methods, such as regression or neural networks-based models.

Moreover, decision trees have demonstrated to be as accurate as such methods in other areas, they support non linearity, and, in fact, the decision trees do a better job at capturing the non-linearity in the data by dividing the space into smaller sub-spaces depending on the questions asked.

Q3. The conclusion of the manuscript might be affected by the fact of the small sample size (78) and the imbalance of the number of samples in each age group, e.g. number of samples in adults (28) is much higher than that in ultra-centenarians (11). The authors need to provide more discussion on this point.

A3. Thanks for this observation. The analyzes were carried out on subgroups belonging to a larger study (Aiello et al., 2021) because these tests are very expensive. So, we changed the conclusion as following:

“..although a note of caution should be added considering the small sample size and the imbalance of the number of samples for each age group.”

Q4. What are the features in predicting the age of the machine learning (ML) model? Did the authors do feature selection? Is the ML model (decision tree) trained on only 78 samples? Would that be a problem in having a very high possibility of overfitting?

A4. In the manuscript, in the first paragraph of section 2.10. (In Material and methods), we wrote “A training and evaluation ratio of 7:3 was used.” That means that the model has been trained using 70% of the total number of samples. That is, about 55 samples for training and the rest for validation. This allows techniques for avoiding overfitting to be used, by augmenting the accuracy with validation data not used during the training process.

Q5. Did the authors try other methods of supervised ML model, e.g. from the simplest linear regression to kernel regression, with and without regularization to compare with the decision tree model used in the manuscript, and see which model would perform the best?

A5. Thank you for this interesting question. We have selected a ML method with a high degree of interpretability, therefore, other classical regression methods have not been tested, since, in fact, those do not offer the same level of interpretability.

Minor comments:

Q6. The authors need to check the texting in the current PDF. For example, in line 219, mojibake was found.

A6. Thanks for the comment. The review was done.

Q7. The quality of the figure, e.g. Figure 4, can clearly be improved.

A7. We improved the quality of Figure 4.

Reviewer 3 Report

First off, I found the concept of this paper to be very interesting. The study of Ultra-Centenarians is very cool and is a rare population to be able to study. Obviously, I would have preferred if the authors had utilized a more unbiased approach (i.e. microrarray or small RNA-seq to analyze these samples, yet I understand that financially this may have not been possible). Nevertheless, I find that the overall paper has conceptual and analysis flaws that require further attention before being suitable for publication. I believe if the authors direct their analysis/focus they can produce a very cool paper using a unique patient population.

First off, the rationale for studying/focusing on these four specific miRNAs is extremely weak. The authors briefly mention in the abstract that these miRNAs are implicated in inflammation and ECs senescence, yet in the introduction there is absolutely no explanation for why these specific miRNAs were investigated in this study!!! There are over >2,500 human miRNAs. Why mir-146a-5p, mir-126-3p, mir-21-5p, and mir-181a-5p???? Without any rationale (and citations) in the beginning of the paper, it will be very difficult for any reader to appreciate the findings in this paper. The authors do eventually get around to discussing the miRNAs in the Discussion, but this is far too late to be introducing any sort of review on the miRNAs, when the entire study is predicated on the analysis of these 4 miRNAs.

Second off, the biggest design flaw in my opinion is that the authors should tie the biology of the miRNAs to their findings in the Ultra-Centenarians. For example, miR-21-5p and miR-126-3p display similar patterns in Ultra-Centenarians. What transcripts are these miRNAs predicted to target? I would ask that the authors to perform a simple target analysis in silico using a miRNA prediction database such as the famous Targetscan or miRWalk (my personal favorite one/more modern approach). Using these predictions, the authors can then try to explain the biology that they are observing, even if it is “speculative”. Failing to explain what these miRNAs may be doing in Ultra-centenarians, will make it difficult to convince any reader of the importance of this study. Even if the authors solely hope to push these miRNAs as aging biomarkers, they must find a way to explain what these miRNAs are doing.

I think Figure 2, may be the most important figure in the paper, and again I find that if the authors perform some sort of target analysis with the miRNAs they can gather some unique insights. For example, miR-21-5p and miR-126-3p display similar patterns (in Ultra-centenarians), while miR-146a-5p and miR-181a-5p display a pretty linear trend. I would ask the authors to analyze what type of genes these miRNAs are targeting. I would take the shared targets from miR-21-5p and miR-126-3p and perform gene ontology analysis to see if anything interesting/aging related appears. Likewise I would do the same with the other two miRNAs. Then the authors could compare the different gene ontology analyses. I think this simple. If the authors are not familiar with these methods, miRWalk can do the analysis for them. For example, I looked at the targets of both miR-21-5p and miR-126-3p, and performed the analysis and found that the two miRNAs target the KEGG: hsa04211_Longevity_regulating_pathway. This is really cool and adds something to the authors paper. A proper analysis done by the authors would make this a very interesting and impactful paper. Furthermore it would sell the merit of using these miRNAs as biomarkers.

Lastly, I am not a machine learning expert, so I do not believe I am qualified to assess the merit of the machine learning design; however I do not believe that this machine learning analysis adds anything important to the paper. Furthermore, I do not believe that the authors have a big enough sample size (I know some statisticians that claim you need at least 1,000 samples per class). Regardless, if the authors want to include this machine learning analysis, they must do a better job at explaining what the machine learning analysis means and how it contributes to the overall study. As it is, the machine learning results are not very surprising. I would like to see how their machine learning analysis would hold up with other aging datasets, or if they were to test the their analysis with other datasets.

Minor Comments

For Figure 3, color coordinate or alter the bars in the HUVECs undergoing senescence, for each age. As it is it’s very hard to tell visualize the data even with the dramatic changes in the miRNAs as the HUVECs go through senescence.

So, our results increase the knowledge about the identification of miRNAs as biomarkers of successful and unsuccessful ageing. Remove the “so”, it flows better to just say “Our results”.

Overall I think the authors have interesting findings, three decent data figures, but as it stands I think it needs a lot of work. No experiments are necessarily needed, just genomic analysis looking at the miRNAs. I hope that the authors take these suggestions, and I think they can paint a very interesting story without the need for wet-lab or intensive dry-lab experiments.

Author Response

REVIEWER 3

First off, I found the concept of this paper to be very interesting. The study of Ultra-Centenarians is very cool and is a rare population to be able to study. Obviously, I would have preferred if the authors had utilized a more unbiased approach (i.e. microrarray or small RNA-seq to analyze these samples, yet I understand that financially this may have not been possible). Nevertheless, I find that the overall paper has conceptual and analysis flaws that require further attention before being suitable for publication. I believe if the authors direct their analysis/focus they can produce a very cool paper using a unique patient population.

We thank the reviewer for the appreciation. Thanks to your suggestions, we will try to improve our paper.

Q1. First off, the rationale for studying/focusing on these four specific miRNAs is extremely weak. The authors briefly mention in the abstract that these miRNAs are implicated in inflammation and ECs senescence, yet in the introduction there is absolutely no explanation for why these specific miRNAs were investigated in this study!!! There are over >2,500 human miRNAs. Why mir-146a-5p, mir-126-3p, mir-21-5p, and mir-181a-5p???? Without any rationale (and citations) in the beginning of the paper, it will be very difficult for any reader to appreciate the findings in this paper. The authors do eventually get around to discussing the miRNAs in the Discussion, but this is far too late to be introducing any sort of review on the miRNAs, when the entire study is predicated on the analysis of these 4 miRNAs.

A1. Thanks to the reviewer for the possibility to improve our manuscript, as also suggested by the reviewer 1 and 2, we modified the introduction and we better explained why we focused our attention on selected miRNAs. Moreover, we added rationale and references, in order to guide the readers in understanding the results.

Q2. Second off, the biggest design flaw in my opinion is that the authors should tie the biology of the miRNAs to their findings in the Ultra-Centenarians. For example, miR-21-5p and miR-126-3p display similar patterns in Ultra-Centenarians. What transcripts are these miRNAs predicted to target? I would ask that the authors to perform a simple target analysis in silico using a miRNA prediction database such as the famous Targetscan or miRWalk (my personal favorite one/more modern approach). Using these predictions, the authors can then try to explain the biology that they are observing, even if it is “speculative”. Failing to explain what these miRNAs may be doing in Ultra-centenarians, will make it difficult to convince any reader of the importance of this study. Even if the authors solely hope to push these miRNAs as aging biomarkers, they must find a way to explain what these miRNAs are doing.

A2. Thanks to the reviewer for the suggestion, we performed an in-silico analysis by miRWalk to identify pathways and genes targeted by our selected miRNAs, in order to clarify the effects of these miRNAs in aging, as shown in Supplemental materials Figures 1,2 and Tables 2-5.

Q3. I think Figure 2, may be the most important figure in the paper, and again I find that if the authors perform some sort of target analysis with the miRNAs they can gather some unique insights. For example, miR-21-5p and miR-126-3p display similar patterns (in Ultra-centenarians), while miR-146a-5p and miR-181a-5p display a pretty linear trend. I would ask the authors to analyze what type of genes these miRNAs are targeting. I would take the shared targets from miR-21-5p and miR-126-3p and perform gene ontology analysis to see if anything interesting/aging related appears. Likewise I would do the same with the other two miRNAs. Then the authors could compare the different gene ontology analyses. I think this simple. If the authors are not familiar with these methods, miRWalk can do the analysis for them. For example, I looked at the targets of both miR-21-5p and miR-126-3p, and performed the analysis and found that the two miRNAs target the KEGG: hsa04211_Longevity_regulating_pathway. This is really cool and adds something to the authors paper. A proper analysis done by the authors would make this a very interesting and impactful paper. Furthermore it would sell the merit of using these miRNAs as biomarkers.

A3. Thanks to the reviewer for the possibility to improve our manuscript, we have analyzed the genes targeted by the 4 selected miRNAs and performed the gene ontology analysis as shown in Supplemental Figures 1,2.

Q4. Lastly, I am not a machine learning expert, so I do not believe I am qualified to assess the merit of the machine learning design; however I do not believe that this machine learning analysis adds anything important to the paper. Furthermore, I do not believe that the authors have a big enough sample size (I know some statisticians that claim you need at least 1,000 samples per class). Regardless, if the authors want to include this machine learning analysis, they must do a better job at explaining what the machine learning analysis means and how it contributes to the overall study. As it is, the machine learning results are not very surprising. I would like to see how their machine learning analysis would hold up with other aging datasets, or if they were to test the their analysis with other datasets.

     A4. We thank the reviewer for this comment. The Introduction paragraph has been rewritten to clarify several issues related to the use of a ML technique for classifying subjects in several ranges of ages, and finding which features are relevant in the clustering process. We added the following paragraph:

“Furthermore, since some specific biological features seem to be age-associated, it was possible to find the relation between these features and people's age, using machine learning (ML) techniques. ML is a branch of Artificial Intelligence focused on imitating the way human beings are capable of learning, taking experimental data and examples as inputs to infer a specific result or output (25). ML often builds mathematical models based on sample data to make predictions without being explicitly programmed for such purposes. In general, ML models are considered as black boxes where inputs are transformed into outputs by combining such inputs with a set of adjusted data obtained as a result of a training process (26). Although ML provides many benefits in different areas of application, the lack of interpretability could be a problem in some circumstances. In the context of this work, the age prediction problem by considering several biological features is also addressed, since it is interesting to study which factors affect the population from young to ultra-centenarians. Finding such relations in an interpretable manner is important, therefore, it is necessary to use ML techniques with a high degree of interpretability. Consequently, a decision tree classifier has been selected for this purpose, since such ML method presents a high rate of accuracy, great robustness and results are very interpretable as long as they are short.”

Please, see as well that:

  1. A) In section 3.5. of the Results, inputs are defined: “(…) For calculation, it was considered miRNAs, C reactive protein, telomeres, paraoxonase (PON), trolox equivalent antioxidant capacity, and malondialdehyde values according to age-ranges”.
  2. B) And the main conclusion for using the model is addressed in the following paragraph found in Section 3, Discussion: “Finally, results from ML pointed out a significant role, in people over 70 years, played by miR-21-5p and miR-181a-5p, in addition to PON and telomere length (46,47).

As it is observed, this ML method is very interpretable and it is possible to get conclusions about which features are significant during the process of classification. This degree of interpretability is not obtained with other methods, such as regression or neural networks-based models.

Minor Comments

Q5. For Figure 3, color coordinate or alter the bars in the HUVECs undergoing senescence, for each age. As it is it’s very hard to tell visualize the data even with the dramatic changes in the miRNAs as the HUVECs go through senescence.

A5. Thanks a lot for the suggestion, we modified the graph in Figure 3.

Q6. So, our results increase the knowledge about the identification of miRNAs as biomarkers of successful and unsuccessful ageing. Remove the “so”, it flows better to just say “Our results”.

A6. Done.

Q7. Overall I think the authors have interesting findings, three decent data figures, but as it stands I think it needs a lot of work. No experiments are necessarily needed, just genomic analysis looking at the miRNAs. I hope that the authors take these suggestions, and I think they can paint a very interesting story without the need for wet-lab or intensive dry-lab experiments.

A7. Thanks. All the requested genomic analysis were done (see paragraph 3.4. Enriched KEGG pathway clustered by validated targets of miR-21-5p, miR-126-3p, miR-146a-5p, and miR-181a-5p and corresponding target genes; see also Supplemental Figures 1,2; Tables 2-5). Moreover, we added this paragraph in the discussion:

Furthermore, the target pathway analysis of miR-126-3p and  miR-21-5p shows  their involvement  also in the nutrient sensing pathway that linked to insulin and insulin growth factor-1, has an important role as “gatekeeper” by balancing the cell response to oxidative stress and nutrient availability. Downstream of this pathway there is the Forkhead box O3 (FOXO3) A transcription factor. Notably, the FOXO family transactivate genes are involved in resistance to oxidative stress, energy metabolism, DNA damage repair, glucose metabolism, autophagy and protection of proteins by chaperones, so favouring survival and longevity (58). Instead, the target pathway analysis of miR-146a-5p and miR-181a-5p shows their involvement also in the sumoylation process. Sumoylation plays critical roles in cellular senescence. Enhancing the global sumoylation or inhibiting desumoylation process seems to promote senescence. In sumoylation-mediated cellular senescence, the p53 and RB proteins are SUMO substrates and have been identified as important molecules in this senescence process (59,60).”

Round 2

Reviewer 1 Report

The article was carefully revised. There are two minor questions.

Q1 The author claims that Qiagen does not provide details regarding assay efficiencies such as primer sequences. In that case, the lot number of the primer should be provided.

Q2 Table2 should be put on the same page.

Author Response

A1. We provided the lot number in the 2.3 paragraph as following (please see in text underlined in yellow).

2.3. TaqMan RT-qPCR miRNA assays

The isolated miRNAs were retro-transcripted using miScript Single Cell qPCR kit (Qiagen, Hilden, Germany), according to the manufacturer’s protocol. The expression levels of miRNAs were evaluated with a SYBR green-based Real-Time quantitative PCR (RT-qPCR), using the Step one plus (Applied Biosystem, Waltham, MA, USA). For the amplification, we used miScript SYBR green PCR kit (Qiagen, Hilden, Germany) according to the manufacturer’s protocol. The 20μl PCR mixture included 2 μl of reverse transcription product, 10 μl of QuantiTect SYBR Green PCR Master Mix, 2μl of miScript Universal Primer, 4μl of RNase-free water and 2μl of Primer Assay specific for each microRNAs (miScript Primer Assays miR-21-5p Lot. N°201803230019, miR-126-3p Lot. N°20150817013s, miR-146a-5p Lot. N°201709120071, miR-181a-5p Lot. N°20161222018  Qiagen, Hilden, Germany). The reaction mixtures were incubated at 95°C for 15 min, followed by 40 amplification cycles of 94°C for 15 s, 55°C for 30 s, and 70°C for 30 s. Triplicate samples and inter-assay controls were used. Therefore, for the normalization of RT-qPCR data, using the 2-DCT method, we used miR-30a (miScript Primer Assay Lot. N° 201709270012 Qiagen, Hilden, Germany). Linear fold changes were calculated and plotted on scatter plots, using Prism (GraphPad Prism Software, San Diego, CA, USA).

A2. Done but a large white space remains on page 8. 

Reviewer 2 Report

The authors have addressed most of my concerns and the manuscript has been largely improved. I can now recommend its publication to Cells.

Author Response

Thank you.